# Evaluation of therapeutic efficacy of baloxavir marboxil against high pathogenicity avian influenza virus infection in duck model

Yo Shimazu[1], Norikazu Isoda[1,2,3,4], Takahiro Hiono[1,2,3,4], Yik Lim Hew[1], Daiki Kobayashi[1], Misato Shibazaki[5], Bao Linh Nguyen[1], Tatsuru Morita[1], Ryo Daniel Obara[6], Mariko Miki[1,6], Hiromi Osaki[6], Takao Shishido[5], Yoshinori Ikenaka[2,7,8,9], Yoshihiro Sakoda[1,2,3,4]*

1 Laboratory of Microbiology, Faculty of Veterinary Medicine, Hokkaido University, Sapporo, Hokkaido, Japan, 2 One Health Research Center, Hokkaido University, Sapporo, Hokkaido, Japan, 3 International Institute for Zoonosis Control, Hokkaido University, Sapporo, Hokkaido, Japan, 4 Institute for Vaccine Research and Development (HU-IVReD), Hokkaido University, Sapporo, Hokkaido, Japan, 5 Laboratory for Drug Discovery and Disease Research, Shionogi & Co., Ltd., Toyonaka, Osaka, Japan, 6 Laboratory for Drug Discovery and Development, Shionogi & Co., Ltd., Toyonaka, Osaka, Japan, 7 Laboratory of Toxicology, Department of Environmental Veterinary Science, Faculty of Veterinary Medicine, Hokkaido University, Sapporo, Hokkaido, Japan, 8 Translational Research Unit, Veterinary Teaching Hospital, Faculty of Veterinary Medicine, Hokkaido University, Sapporo, Hokkaido, Japan, 9 Water Research Group, Unit for Environmental Sciences and Management, North-West University, Potchefstroom, South Africa

* sakoda@vetmed.hokudai.ac.jp

## Abstract

High pathogenicity avian influenza virus (HPAIV) infections have been frequently reported in wild birds since 2020. Because HPAIV infection has occasionally caused outbreaks in captive rare birds, treatment using antiviral drugs has been considered from the perspective of conservation medicine. In this study, the therapeutic efficacy of baloxavir marboxil (BXM) was evaluated using a duck model to support the establishment of post-infection strategies for captive avian species. Sixteen four-week-old ducks were divided into four groups and intranasally inoculated with the HPAIV strain, A/crow/Hokkaido/0103B065/2022 (H5N1). BXM was administered orally once daily at doses of 12.5, 2.5, 0.5, or 0 mg/kg to each of the four groups from 2 to 6 days post-infection. Blood samples were collected at 2, 8, and 24 hours after the initial BXM administration to measure the plasma concentrations of its active form, baloxavir acid (BXA). All ducks were monitored until 14 days post-infection, and oral and cloacal swabs were collected for virus recovery. All eight ducks administered either 12.5 or 2.5 mg/kg of BXM survived, demonstrating a significant reduction in virus recovery compared with the 0 mg/kg group. To further characterize the relationship between BXA exposure and virus shedding, an additional experiment was conducted in which ducks received a single oral administration of BXM at doses of 7.3, 4.3, or 0.5 mg/kg at 2 days post-infection. Pharmacokinetic/pharmacodynamic analysis of BXA across all dose groups showed that reductions in virus shedding from oral swabs were positively correlated with BXA exposure, and $AUC_{0-24hr}$ exhibited the strongest association. These findings demonstrate that BXM administration within 48 hours post-HPAIV

**Data availability statement:** All relevant data are within the manuscript and its Supporting Information files.

**Funding:** This research was conducted as part of a collaborative research project between Hokkaido University and Shionogi & Co., Ltd. It was supported by the Environment Research and Technology Development Fund [JPMEERF20254004] of Environmental Restoration and Conservation Agency provided by Ministry of the Environment of Japan. Additionally, this work was partially supported by the Japan Agency for Medical Research and Development (AMED) [grant numbers JP223fa627005 and JP24wm000125008] and by the World-Leading Innovative and Smart Education Program (1801) from the Ministry of Education, Culture, Sports, Science and Technology, Japan. It was also supported by the Japan Science and Technology Agency SPRING grant [number JPMJSP2119]. The funders had no role in study design, data collection and analysis, decision to publish, or preparation of the manuscript.

**Competing interests:** I have read the journal's policy and the authors of this manuscript have the following competing interests: M.S., R.D.O, H.O., and T.S. are employees of Shionogi & Co., Ltd. M.M. is a former employee of Shionogi & Co., Ltd. These affiliations did not influence the study design, data collection, analysis, or interpretation. The remaining authors declare no competing interests.

infection in ducks effectively reduces mortality and virus shedding. Comparison of pharmacokinetic parameters may facilitate the estimation of optimal BXM dosing options for captive avian species.

High pathogenicity avian influenza viruses (HPAIVs), classified as members of influenza A viruses (*Alphainfluenzavirus influenzae*), cause fatal disease in various bird species [1]. In recent years, these viruses have spread widely among wild bird populations, and their impact is now recognized on a global scale [2]. Since 2020, infections of HPAIVs belonging to clade 2.3.4.4b have been continuously reported in various regions, including Europe, Asia, Africa, the Americas, and Antarctica [3,4]. These epidemics have shifted from a primarily seasonal to a year-round endemic situation in Europe. Under these circumstances, HPAIV outbreaks have also occurred among captive birds in zoos, resulting in the death of endangered avian species [5,6]. Vaccination has been implemented in some zoos during domestic outbreaks of HPAIVs in several countries, including Singapore, the Netherlands, Spain, Denmark, and the United States [7–11], but the use of vaccines in captive birds is not available in several countries, including Japan. In these countries, infection control relies mainly on enhanced biosecurity measures, and once HPAIV infection is detected in captive birds, infected and contact birds are recommended to be culled to prevent further spread of infection and eliminate the virus [12]. Nevertheless, from the perspective of conservation medicine, therapeutic options for rare birds should be explored. In zoos, birds are often housed in open cages to support animal welfare, placing limitations on biosecurity. Under these circumstances, both prophylactic and therapeutic countermeasures are important. Currently, there is no established treatment protocol for HPAIV infection in captive birds, and the development of effective antiviral therapies is urgently needed.

Several antiviral drugs targeting the viral neuraminidase (NA) and polymerase acidic (PA) protein are widely used for treatment of human influenza virus infection. The NA, a surface glycoprotein of the influenza virus, exhibits sialidase activity and facilitates virus release from infected cells [13]. The NA inhibitors oseltamivir, zanamivir, and peramivir have been evaluated for their antiviral effects against HPAIVs in chicken models [14,15]. However, none of these drugs completely prevented the onset of disease and death. Even when administered simultaneously with HPAIV challenge, they reduced mortality by only about 80% (oseltamivir), 0% (zanamivir), and 75% (peramivir). NA inhibitors suppress viral release in the late stages of infection, but as they do not inhibit viral replication, their effect is thought to be limited. The PA, a subunit of the viral RNA-dependent RNA polymerase complex of the influenza virus, is involved in cap-snatching, by acquiring capped RNA fragments from host mRNAs via its endonuclease activity [16]. PA inhibitors show higher antiviral efficacy than NA inhibitors by suppressing viral replication in the early stages of infection [17]. Baloxavir marboxil (BXM) is a PA inhibitor that has been approved for clinical use in humans, but not in animals. Simultaneous administration of BXM at 2.5 mg/kg or higher to chickens

inoculated with HPAIV demonstrated a prophylactic effect in suppressing disease onset [15]. Assuming an initial infection case in zoos, treatment would be initiated after captive birds showed clinical symptoms and were diagnosed. The previous study also evaluated the efficacy of BXM following post-infection administration in a chicken model; however, even administering 20 mg/kg of BXM at 12-hour intervals from 24 hours post-infection, all individuals died, and no efficacy of post-infection administration was observed [15]. This result is likely due to the extremely rapid disease progression of HPAIVs in chickens, which typically results in death within 2–3 days post-infection [18], thus heavily limiting the time window for therapy in the chicken model. However, infection with HPAIVs in some non-galliform species, such as sparrows, crows, herons, white-tailed eagles, and wild waterfowls, often results in slower progression (4–7 days from infection to death) or is even sublethal [19–22]. Therefore, an avian model with slower disease progression is needed for evaluating the post-infection efficacy of BXM administration and for developing treatment strategies applicable to captive avian species.

In this study, the ducks were utilized as the model animal species, because their slower progression of clinical signs in HPAIV infection compared to chickens enables more accurate extrapolation of outcomes to captive avian species [20]. BXM was administered to infected ducks, and their survival rates, clinical symptoms, and virus shedding were monitored over time. Based on the results, an effective dose and optimal BXM administration schedule were proposed. Furthermore, pharmacokinetic parameters in plasma samples collected after BXM administration were analyzed to investigate their association with antiviral effects. The findings of this study provide a foundational reference for establishing practical treatment strategies for HPAIV infection in captive avian species.

## Materials and methods

### Viruses

Three HPAIVs, A/crow/Hokkaido/0103B065/2022 (H5N1) (Cr/Hok/22) belonging to clade 2.3.4.4b [23], A/black swan/Akita/1/2016 (H5N6) (Bs/Aki/16) belonging to clade 2.3.4.4e [24], and A/whooper swan/Hokkaido/4/2011 (H5N1) belonging to clade 2.3.2.1 (Ws/Hok/11) [25], were isolated from a dead large-billed crow (*Corvus macrorhynchos*) collected from an urban garden, a captive black swan (*Cygnus atratus*) bred at a zoo, and a wild whooper swan (*Cygnus cygnus*) recovered near its resting waters, respectively. The HPAIVs were propagated in 10-day-old embryonated chicken eggs for 44 hours at 35°C. The collected allantoic fluids were stored as the virus stocks for experimental infection.

### Animals

Thirty-seven four-week-old Cherry Valley ducks (*Anas platyrhynchos* var. *domesticus*, 1.4–2.1 kg body weight, Anshin-Seisan Farm, Horonobe, Hokkaido, Japan) were used in this study. Before starting the experiment, it was confirmed that none of the ducks possessed antibodies to influenza A virus, using the IDEXX Influenza A Ab Test (IDEXX Laboratories, Inc., Westbrook, ME, USA), an enzyme-linked immunosorbent assay (ELISA) kit. All ducks were provided with environmental enrichment appropriate for the species and maintained under conditions that complied with institutional animal welfare guidelines.

### Cells

Madin—Darby canine kidney (MDCK) cells were maintained in minimum essential medium (MEM) (Shimadzu Diagnostics Corp., Tokyo, Japan) supplemented with 0.3 mg/mL L-glutamine (FUJIFILM Wako Pure Chemical Corp., Osaka, Japan), 5% fetal bovine serum (Thermo Fisher Scientific Inc., Waltham, MA, USA), 100 U/mL penicillin G (Meiji Seika Pharma Co., Ltd., Tokyo, Japan), 0.1 mg/mL streptomycin (Meiji Seika Pharma), and 8.0 μg/mL gentamicin (MSD, Rahway, NJ, USA).

### Experimental infection of ducks

As a preliminary test to select the virus strain for the subsequent experiment, nine ducks were divided into three groups and intranasally challenged with $10^{6.0}$ times of 50% egg infectious doses ($EID_{50}$) of Cr/Hok/22, Bs/Aki/16, and Ws/Hok/11.

The ducks were monitored at least twice daily. During the first daily observation, clinical scores were determined by applying the criteria used for the calculation of the intravenous pathogenicity index according to the World Organisation for Animal Health manual [26]. Briefly, birds were scored 0 if healthy, 1 if sick, 2 if severely sick, and 3 if dead. Birds were considered sick if one of the following signs was observed, and severely sick if more than one was observed: respiratory involvement, depression, diarrhea, cyanosis of the feet or mucosa, edema of the face or head, and nervous signs. During the second observation, ducks were assessed specifically for the presence of severe clinical signs indicating that pre-defined humane endpoints had been reached. When severe neurological signs or respiratory distress were observed, the ducks were deemed to have reached the humane endpoint and euthanized within 8 hours thereafter. From the following day, they were scored as 3. Euthanasia was performed by intravenous administration of an overdose of thiopental, following the institutionally approved method to minimize suffering. The ducks were kept for 14 days post-infection (dpi), and all remaining animals were humanely euthanized at the end of the experimental period in accordance with the institutionally approved protocol. No analgesic or palliative interventions were applied during the study to allow accurate evaluation of lethal pathogenicity, with humane euthanasia performed when predefined humane endpoints were reached.

Sixteen ducks were divided into four groups, and all ducks were intranasally challenged with $10^{6.0}$ $EID_{50}$ of Cr/Hok/22. BXM (20 mg tablets, Shionogi & Co., Ltd., Osaka, Japan) was ground and suspended in saline (Otsuka Pharmaceutical Factory, Inc., Tokushima, Japan). BXM suspension was administered orally to three groups of ducks once daily at doses of 12.5, 2.5, or 0.5 mg/kg, from 2 to 6 dpi. Saline was administered to the 0 mg/kg group during the study period. The five-day administration regimen was designed considering the shorter elimination half-life of the drug in ducks compared with humans, to maintain adequate plasma exposure during treatment [15]. Clinical manifestations were observed at least twice daily in the same way. Oral and cloacal swabs of the challenged birds were collected at 0–7, 9, 11, and 14 dpi and suspended in 2 mL of viral transport medium consisting of MEM (Shimadzu Diagnostics) containing 10,000 U/mL of penicillin G (Meiji Seika Pharma), 10 mg/mL streptomycin (Meiji Seika Pharma), 0.3 mg/mL gentamicin (MSD), 0.05 mg/mL nystatin (Meiji Seika Pharma), and 0.5% of bovine serum albumin fraction V (Roche Diagnostics GmbH, Mannheim, Germany). Viral titers were calculated by the method of Reed and Muench [27] and expressed as the 50% tissue culture infectious dose ($TCID_{50}$) per milliliter of swab suspension. The ducks were kept for 14 dpi, and all remaining animals were humanely euthanized at the end of the experimental period in accordance with the institutionally approved protocol. No analgesic or palliative interventions were applied during the study to allow accurate evaluation of lethal pathogenicity and antiviral efficacy, with humane euthanasia performed when predefined humane endpoints were reached. To measure the plasma concentration of baloxavir acid (BXA), the active form of BXM, 230 µL blood samples were collected from ducks administered BXM at 12.5 and 2.5 mg/kg at 2, 8, and 24 hours post-initial dose (spanning from 2 to 3 dpi), before the fifth dose, and 2, 8, and 24 hours post-fifth dose (spanning from 6 to 7 dpi). The collected blood samples were thoroughly mixed with 13.8 µL of an enzyme inhibitor/anticoagulant mixture (60 mmol/L dichlorvos [FUJIFILM Wako Pure Chemical]/sodium heparin [Nipro Corp., Osaka, Japan]), and centrifuged at 1900 g for 15 minutes to obtain plasma.

To further investigate the relationship between BXM dose and virus shedding, an additional experiment was conducted. Twelve ducks were divided into four groups and intranasally challenged with $10^{6.0}$ $EID_{50}$ of Cr/Hok/22. BXM was ground and suspended in saline. Three groups received a single oral dose of BXM at 7.3, 4.3, or 0.5 mg/kg at 2 dpi, while the control group received saline. The 7.3 and 4.3 mg/kg doses were selected as intermediate levels to further characterize the dose–response relationship between BXM administration and virus shedding. The 0.5 mg/kg dose was included to enable pharmacokinetic sampling that was not feasible in the previous experiment due to technical limitations. Oral and cloacal swabs were collected at 2 and 3 dpi, and virus titers in the swab samples were determined as described above. In the BXM-treated ducks, 230 µL of blood was collected at 2, 8, and 24 hours post-administration, and plasma was prepared as previously described. All ducks were humanely euthanized at 3 dpi in accordance with the institutionally approved protocol. No analgesic or palliative interventions were applied during the study to allow for an accurate evaluation of antiviral efficacy, with humane euthanasia performed when predefined humane endpoints were reached.

All animal experiments, including the predefined humane endpoints and the use of a lethal infection model without analgesic or palliative interventions, were reviewed and approved by the Institutional Animal Care and Use Committee of the Faculty of Veterinary Medicine, Hokkaido University (approval no. 23–0052, approved on 30 March 2023), and conducted in accordance with its guidelines. The Faculty of Veterinary Medicine at Hokkaido University has maintained accreditation from the Association for Assessment and Accreditation of Laboratory Animal Care International since 2007. All personnel involved in the animal experiments had completed the institutionally required education and training programs for animal experimentation and animal welfare.

### Quantitative RT-PCR

Viral RNAs were extracted from oral and cloacal swab samples using the QIAamp viral RNA Mini Kit (QIAGEN GmbH, Hilden, Germany) according to the manufacturer's instructions and stored at –80°C until further processing. The presence of the matrix gene was investigated by quantitative reverse transcription-polymerase chain reaction (RT-qPCR) using THUNDERBIRD Probe One-step qRT-PCR Kit (Toyobo Co., Ltd., Osaka, Japan) on a LightCycler 480 System (Roche Diagnostics) with primer sets as described by Heine et al. [28].

### Pharmacokinetic analysis

The plasma concentration of BXA was determined using liquid chromatography–tandem mass spectrometry (Agilent 6495B, Agilent Technologies, Inc., Santa Clara, CA, USA). The lower limit of quantification for BXA was 4 ng/mL. The pharmacokinetic parameters including maximum plasma concentration ($C_{max}$), area under the plasma concentration–time curve from 0 to 24 hours ($AUC_{0-24hr}$), and elimination rate constant ($k_{el}$) were determined using the linear trapezoidal method, and plasma half-life ($T_{1/2}$) was calculated from $k_{el}$.

### Pharmacokinetic/pharmacodynamic analysis

To assess the relationship between pharmacokinetic parameters and antiviral efficacy, the independent variables (x-axis) were set as the pharmacokinetic parameters $AUC_{0-24hr}$, $C_{max}$, and plasma concentration of BXA at 24 hours post-administration ($C_{24hr}$). The dependent variable (y-axis) was defined as the difference in viral titers between 2 and 3 dpi. Logarithmic trend lines were fitted using Excel (Microsoft, Redmond, WA, USA), and the corresponding correlation coefficients ($R^2$ values) were calculated.

### Statistical analysis

The mean and standard deviation of values for clinical score, virus titer, and plasma concentration of BXA were calculated using Excel (Microsoft). A one-sided Student's *t*-test was performed to compare each dosage group with the 0 mg/kg group. For virus titers below the detection limit, a value of $10^{0.8}$ $TCID_{50}$ was assigned for statistical calculation. The survival rate was compared among groups using a log-rank test in OriginPro 2025 (OriginLab Corp., Northampton, MA, USA). The correlation between pharmacokinetic parameters and reduction of virus titer was assessed using test of no correlation using Excel (Microsoft). When the *p*-value was less than 0.05, the difference was regarded as statistically significant.

## Results

### Antiviral effects of baloxavir marboxil against high pathogenicity avian influenza virus infection in ducks

All birds inoculated with Cr/Hok/22 died within 11 dpi (S1A Fig). Of those inoculated with Bs/Aki/16, one bird died at 9 dpi and two birds survived. All three birds inoculated with Ws/Hok/11 survived. Mean clinical scores were relatively high in birds inoculated with Cr/Hok/22 (S1B Fig). To enable precise evaluation of BXM efficacy, Cr/Hok/22 was selected for

subsequent experiments because of its high pathogenicity in ducks. In this preliminary experiment, all ducks that died were found dead.

Four ducks per group were intranasally inoculated with $10^{6.0}$ EID$_{50}$ of Cr/Hok/22. In the 0 mg/kg group, all ducks died or reached the humane endpoint within 7 dpi (Fig 1A). In the 0.5 mg/kg treatment group, one of the four treated birds died at 5 dpi, and the other three birds survived for the experimental period, indicating an apparent improvement in survival rate ($p = 0.056$). None of the birds administered 12.5 or 2.5 mg/kg died after virus challenge, demonstrating complete suppression of mortality in these groups compared to the 0 mg/kg group ($p = 0.007$). No clinical symptoms (or only transient symptoms followed by recovery) were observed in any of the BXM-treated ducks, except for one in the 0.5 mg/kg group that died at 5 dpi and one in the 12.5 mg/kg group that exhibited leg paralysis until the end of the study despite recovery from acute symptoms (Fig 1B, S1 Table). Among the ducks in the 0 mg/kg group, two reached predefined humane endpoints and were euthanized, whereas the remaining ducks were found dead.

The virus titers recovered from oral swabs of the 0.5 mg/kg-treated ducks on 2 days post-initial dose (dpid) (equal to 4 dpi) were significantly lower than those of the 0 mg/kg group on that day ($p = 0.041$) (Fig 2A). In the 12.5 and 2.5 mg/kg groups, the titers of virus recovered from oral swabs decreased significantly from 1 dpid (3 dpi) compared with the 0 mg/kg group ($p < 0.001$ [12.5 mg/kg], $p = 0.001$ [2.5 mg/kg]), and were below the detection limit from 3 dpid (5 dpi) onward. In

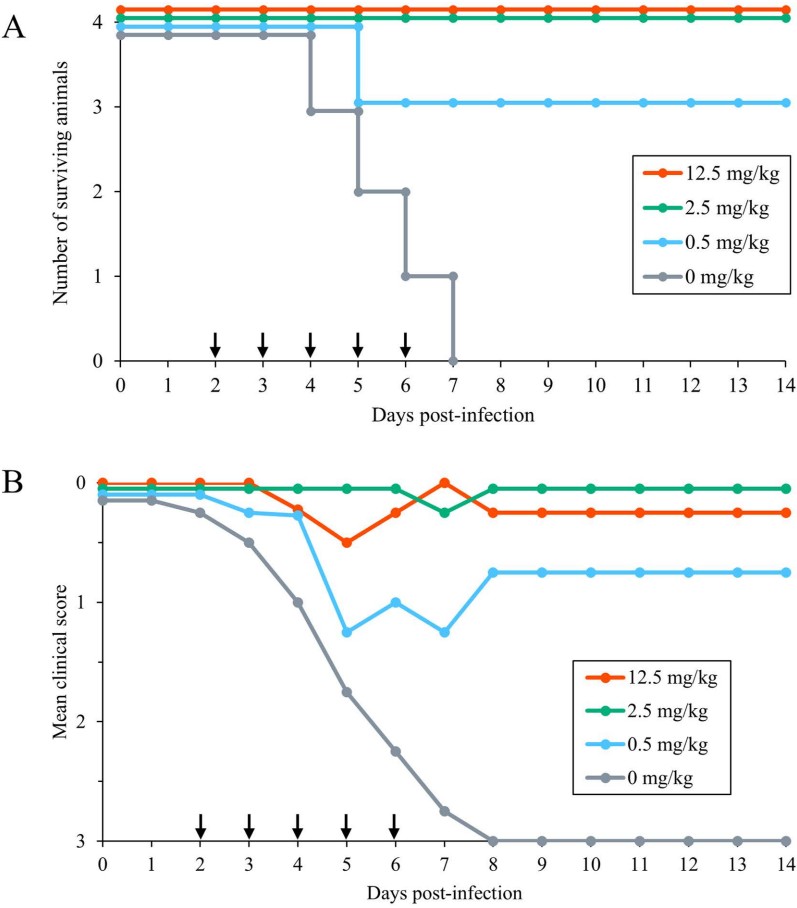

**Fig 1. Efficacy of post-infection administration of baloxavir marboxil (BXM) in ducks infected with A/crow/Hokkaido/0103B065/2022 (H5N1).** (A) Survival curves. (B) Mean clinical scores. BXM was administered orally at 12.5, 2.5, 0.5, or 0 mg/kg once daily on days 2–6 post-infection (arrows).

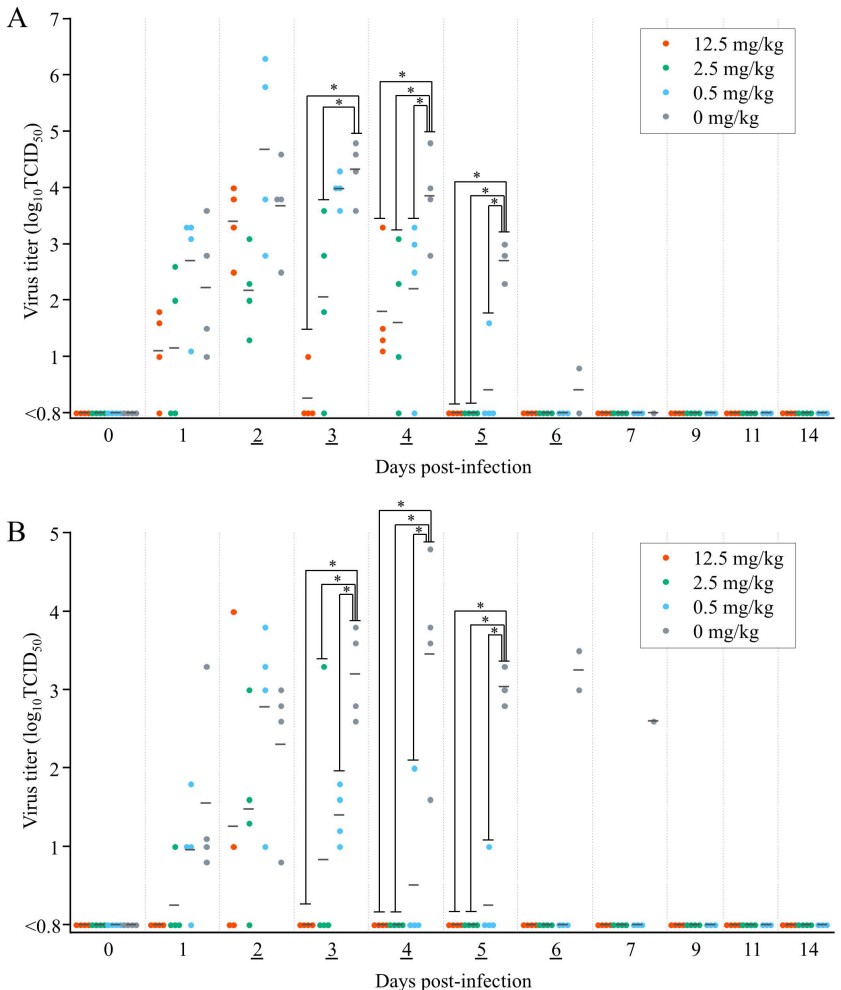

**Fig 2. Virus titer of (A) oral and (B) cloacal samples in ducks infected with A/crow/Hokkaido/0103B065/2022 (H5N1).** Baloxavir marboxil was administered orally at 12.5, 2.5, 0.5, or 0 mg/kg once daily on days 2–6 post-infection (underlined). Bars represent the mean values. *$p < 0.05$ versus 0 mg/kg administration group.

cloacal swabs, the virus titers of the ducks treated with all three doses were significantly lower than those of the 0 mg/kg group at 1 dpid (3 dpi) ($p < 0.001$ [0.5 mg/kg], $p = 0.021$ [2.5 mg/kg], $p = 0.001$ [12.5 mg/kg]). Furthermore, from 2 dpid (4 dpi), all virus titers of ducks administered 12.5 and 2.5 mg/kg were below the detection limit of $10^{0.8}$ TCID$_{50}$.

Although the recovery of infectious virus from swab samples was limited to 2 or 3 dpid, viral genomes were continuously detected until 4, 5, and 5 dpid in oral swabs of ducks administered BXM at 12.5, 2.5, and 0.5 mg/kg, respectively (S2 Table).

### Pharmacokinetic analysis of baloxavir acid in ducks infected with high pathogenicity avian influenza virus

To evaluate the pharmacokinetics of BXA, the active form of BXM, plasma samples were collected at 2, 8, and 24 hours post-initial dose of BXM (spanning from 2 to 3 dpi) and at 0, 2, 8, and 24 hours post-fifth administration of BXM (spanning from 6 to 7 dpi) in the 12.5 and 2.5 mg/kg groups, and the plasma concentration of BXA in each sample was analyzed. In addition, an independent pharmacokinetic study was conducted in nine ducks that were administered BXM at doses of 7.3, 4.3, or 0.5 mg/kg. Plasma samples were collected at 2, 8, and 24 hours post-administration to assess BXA exposure. No clinical signs were observed in any of the animals throughout the study period.

After the single administration of BXM at 12.5, 7.3, 4.3, 2.5, or 0.5 mg/kg, the plasma concentration of BXA reached 1784, 541, 641, 285, or 51 ng/mL at 2 hours post-administration, and decreased to 130, 96, 58, 26, or 9.0 ng/mL at 24 hours post-administration, respectively (Fig 3, S3,S4 Tables). After the fifth administration of BXM at 12.5 or 2.5 mg/kg, the plasma concentration of BXA increased to 876 and 145 ng/mL at 2 hours post-administration and decreased to 18 and 6 ng/mL at 24 hours post-administration, respectively. Based on these results, pharmacokinetic parameters were calculated. Parameters reflecting BXA plasma exposure, including $AUC_{0-24hr}$ and $C_{24hr}$, increased in a dose-dependent manner, while $C_{max}$ did not increase proportionally with dose. In addition, $AUC_{0-24hr}$, $C_{max}$, and $C_{24hr}$ of BXA after the fifth administration were lower than those observed after the initial administration during the 5-day repeated dosing period.

## Pharmacokinetic/pharmacodynamic analysis of baloxavir acid in ducks infected with high pathogenicity avian influenza virus

Correlations between pharmacokinetic parameters from the 12.5, 7.3, 4.3, 2.5, and 0.5 mg/kg groups and changes in virus recovery before and after treatment were analyzed. For each type of swab, the difference in virus titers between the day of initial BXM administration (2 dpi) and 1 dpid (3 dpi) was calculated. In oral swabs, mean reductions in virus titers were 3.2, 0.8, 1.3, 0.1, –0.7 $\log_{10}$ $TCID_{50}$/mL in the 12.5, 7.3, 4.3, 2.5, and 0.5 mg/kg groups, respectively. These values correlated

A

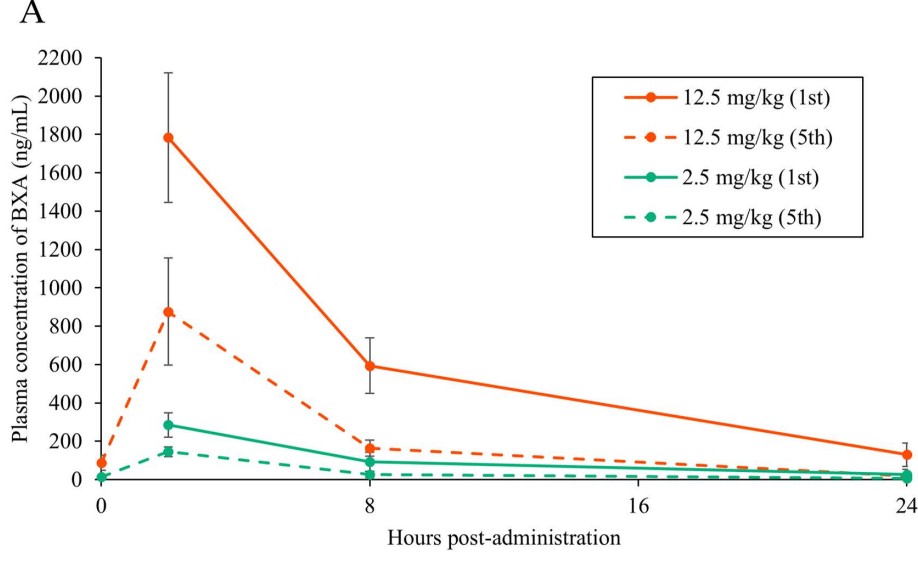

B

| Time of administration | Dose (mg/kg) | $AUC_{0-24hr}$ (ng·hr/mL) | $C_{max}$ (ng/mL) | $C_{24hr}$ (ng/mL) | $T_{1/2}$ (hr) |
|---|---|---|---|---|---|
| 1 | 12.5 | 14,170 ± 3,357 | 1,784 ± 338 | 130 ± 61 | 5.9 ± 0.8 |
|  | 2.5 | 2,374 ± 608 | 285 ± 64 | 26 ± 27 | 7.5 ± 4.6 |
| 5 | 12.5 | 5,546 ± 1566 | 876 ± 280 | 18 ± 4 | 4.2 ± 0.5 |
|  | 2.5 | 940 ± 265 | 145 ± 26 | 6 ± 3 | 5.2 ± 0.4 |

**Fig 3. (A) Mean plasma concentration of baloxavir acid (BXA) after administration of baloxavir marboxil (BXM) at 12.5 and 2.5 mg/kg. Data represent the mean ± standard deviation of four ducks. (B) Mean ± standard deviation pharmacokinetic parameters of BXA in the plasma of ducks administered BXM at 12.5 or 2.5 mg/kg.**

with $AUC_{0-24hr}$, $C_{max}$, and $C_{24hr}$, with $R^2$ values of 0.703 ($p < 0.001$), 0.645 ($p < 0.001$), and 0.581 ($p < 0.001$), respectively (Fig 4). In cloacal swabs, mean reductions in virus titers were 1.3, 0, 0.6, 0.7, and 0.5 $log_{10}$ $TCID_{50}$/mL in the 12.5, 7.3, 4.3, 2.5, and 0.5 mg/kg groups, and $R^2$ values for $AUC_{0-24hr}$, $C_{max}$, and $C_{24hr}$ were 0.099 ($p = 0.170$), 0.037 ($p = 0.213$), 0.182 ($p = 0.339$), respectively. This finding suggests that the reduction of virus shedding in oral swabs is most strongly associated with $AUC_{0-24hr}$. In contrast, in cloacal swabs, no clear correlation was observed between the reduction in virus shedding and pharmacokinetic parameters.

## Discussion

The application of human anti-influenza drugs for the treatment of HPAIV infection in birds has been considered [14,15]. A previous study reported that prophylactic administration of oseltamivir, zanamivir, and peramivir only partially reduced mortality in chickens infected with HPAIV, whereas BXM suppressed the onset of HPAIV infection at a dose of 2.5 mg/kg or higher when administered concurrently with virus challenge [15]. In the present study, the ducks were used as the model species because disease manifestations typically last for 4–7 days after HPAIV inoculation [20], similar to those of HPAIV infection in wild birds. The results of the present study demonstrated the post-infection treatment efficacy of BXM against HPAIV infection in ducks, as it markedly decreased mortality with a significant reduction in virus shedding. Due to

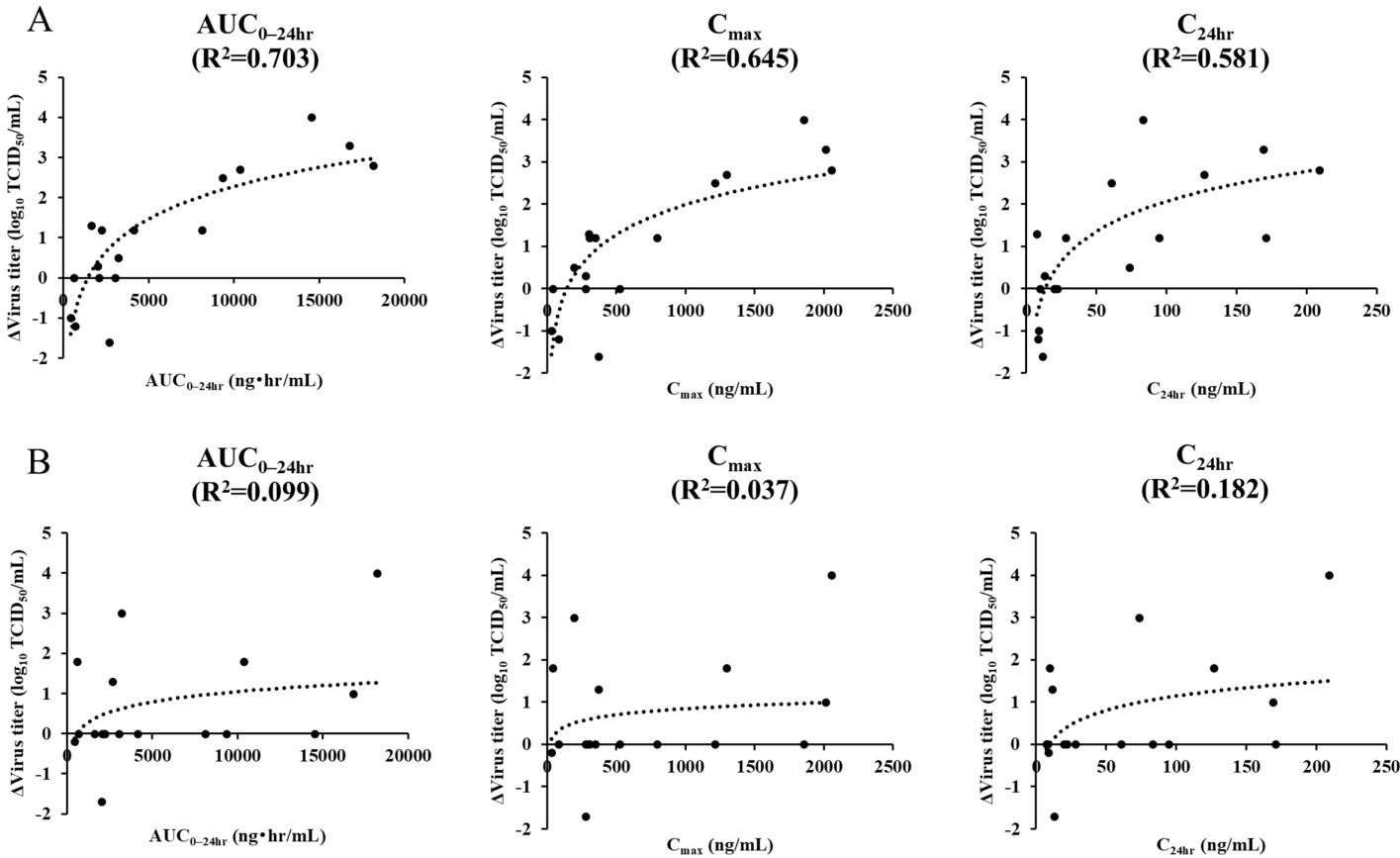

**Fig 4. Estimated linear curves between each pharmacokinetic parameter and the reduction of virus titers in oral (A) and cloacal (B) swabs between days 2 and 3 post-infection.**

their slower progression of clinical symptoms, the duck model was more suitable than the chicken model for evaluating the efficacy of post-infection administration of BXM.

An improvement in survival rate was observed among ducks administered 0.5 mg/kg of BXM following HPAIV challenge. Moreover, all birds survived at a dose of 2.5 mg/kg, confirming a dose-dependent effect on clinical onset with delayed administration. However, one bird in the 12.5 mg/kg dose group developed leg paralysis. Previous studies have shown that viruses can be detected in duck brains at 2–3 days post HPAIV infection and may cause neurological signs [29,30], implying that the virus might have invaded the duck's nervous system before treatment in the present study. Neural tissue damage is known to be irreversible [31]; thus, delayed treatment reduces the probability of complete clinical recovery. It is therefore necessary to administer BXM soon after confirming virus infection in captive birds in situations with high risk of HPAIV infection.

The titers of virus recovered from oral swabs of ducks administered 0.5 mg/kg significantly decreased from 2 dpid ($p = 0.012$). Virus titers in oral swabs were below the detection limit from 4 dpid onward. A significant reduction in virus titers in oral swabs was observed in the 12.5 and 2.5 mg/kg groups ($p = 0.001$, $p < 0.001$) from 1 dpid, and the titers were below the detection limit from 3 dpid onward. These findings suggest that administration at doses of 2.5 mg/kg or higher rapidly suppresses virus shedding. In addition, oral swabs showed higher virus titers than cloacal swabs, indicating that oral swabs may be more appropriate for evaluating the inhibitory effect of BXM on virus shedding.

In the 12.5 mg/kg group, virus shedding was suppressed at 1 dpid, whereas virus shedding re-emerged at 2 dpid, suggesting the potential involvement of drug-resistant viruses. Previous studies also demonstrated that drug-resistant viruses can be rapidly selected following delayed administration [32]. The BXM-resistant variant of H5 HPAIV harboring the PA/I38T substitution, the most frequently emerging mutation under BXM treatment, exhibited up to a 48-fold increase in the 90% effective concentration compared with the susceptible variant [33], corresponding to a BXA concentration of 25.8 ng/mL. The concentration at 2 hours post-fifth administration transiently exceeded this value, suggesting that BXM could exert antiviral activity against resistant viruses when administered at doses greater than 2.5 mg/kg. In the present study, administration of BXM at a sufficient dose for five consecutive days led to a marked reduction in viral shedding in infected ducks. These results highlight the importance of predefining both the dose and duration of administration in the target avian species and adhering strictly to the established treatment protocol. Such optimization is important for anticipating and mitigating the potential risk of selecting resistant variants during the treatment of rare birds infected with HPAIV.

In the present study, viral genome detection was performed by RT-qPCR in parallel with monitoring of infectious virus shedding. The persistence of viral genome detection, even after infectious virus shedding disappeared, is consistent with previous reports of long-term persistence of non-infectious viral genome shedding in ducks [34]. Therefore, while RT-qPCR is effective for detecting early infection, it may not be appropriate for determining the therapeutic efficacy of BXM.

Pharmacokinetic analysis confirmed that BXA appeared in plasma after oral administration of BXM, with the exposure increasing in a dose-dependent manner. However, a decrease in the plasma concentration of BXA was observed after five days of BXM administration, probably due to the induction of metabolic enzymes that accelerate drug clearance. Consequently, this reduction in drug concentration does not support continuous administration of BXM as a prophylactic measure during the HPAIV endemic season. A previous study suggested $C_{24hr}$, the plasma concentration of BXA at 24 hours post-administration, as a surrogate marker for the antiviral effect of BXM using a mouse model of infection with the A/WSN/33 (H1N1) strain [35]. However, this parameter was derived from experiments using a mouse-adapted neurotropic strain and may not be directly applicable to HPAIV infection in avian models, which typically result in systemic infection. In the mouse model, $C_{24hr}$ correlated strongly with the reduction in virus shedding rather than $AUC_{0-24hr}$ and $C_{max}$, indicating that maintaining effective concentrations is important. However, the present study found that $AUC_{0-24hr}$ showed a higher correlation with the suppression of virus shedding in oral swabs than $C_{max}$ or $C_{24hr}$, indicating that higher exposure to BXA may correlate with anti-HPAIV efficacy in birds. Therefore, in avian HPAIV infection, $AUC_{0-24hr}$ appears to be a useful surrogate marker for suppression of virus shedding. Furthermore, achieving an $AUC_{0-24hr}$ of 2374 ng·hr/

mL, which was observed at a dose of 2.5 mg/kg and was associated with complete prevention of mortality and rapid suppression of virus shedding, may be important when applying BXM treatment to other avian species. Pharmacokinetic analyses of BXM and BXA have been conducted in several avian species to facilitate extrapolation of dosing regimens. In a previous study using Okinawa rails (*Hypotaenidia okinawae*), dosing regimens were determined based on the $C_{24hr}$ value obtained from pharmacokinetic analysis [36]. Re-evaluation of the regimens based on $AUC_{0-24hr}$ may provide a more reasonable protocol.

## Conclusion

This study highlights the potential of BXM as a post-infection treatment against HPAIV infection in birds, identifying a 48-hour therapeutic time window and an effective dose of 2.5 mg/kg for reducing mortality and virus shedding. The efficacy of post-infection administration could be associated with the pharmacokinetics of BXA in plasma, particularly $AUC_{0-24hr}$, which can serve as an effective surrogate marker. Comparing the pharmacokinetics of BXA in ducks with those in captive birds may facilitate the establishment of effective BXM treatment protocols for HPAIV in different avian species. Although BXM administration to ducks infected with HPAIV may have resulted in the emergence of resistant viruses, five consecutive days of BXM administration suppressed virus shedding. Collectively, these findings provide fundamental data to inform the rational design of BXM administration protocols for treating HPAIV infection in captive avian species and ultimately contribute to avian conservation efforts.

## Supporting information

**S1 Fig. (A) Survival rates and (B) mean clinical scores of ducks infected with A/crow/Hokkaido/0103B065/2022 (H5N1) (Cr/Hok/22), A/black swan/Akita/1/2016 (H5N6) (Bs/Aki/16), and A/whooper swan/Hokkaido/4/2011 (H5N1) (Ws/Hok/11).**
(TIF)

**S1 Table. Clinical symptoms and hemagglutination inhibition (HI) titers in serum of ducks infected with A/crow/Hokkaido/0103B065/2022 (H5N1) followed by administration of baloxavir marboxil at 12.5, 2.5, 0.5, or 0 mg/kg.**
(XLSX)

**S2 Table. Virus titers and Cp values of oral and cloacal swabs from ducks infected with A/crow/Hokkaido/0103B065/2022 (H5N1) followed by administration of baloxavir marboxil at 12.5, 2.5, 0.5, or 0 mg/kg.**
(XLSX)

**S3 Table. Plasma concentration of baloxavir acid (BXA) after first and fifth administrations of baloxavir marboxil at 12.5 and 2.5 mg/kg.**
(XLSX)

**S4 Table. Virus titers in oral and cloacal swabs, plasma concentrations of baloxavir acid (BXA), and pharmacokinetic parameters of BXA in ducks infected with A/crow/Hokkaido/0103B065/2022 (H5N1) and treated with baloxavir marboxil at 7.3, 4.3, or 0.5 mg/kg.**
(XLSX)

## Acknowledgments

We would like to express our sincere gratitude to Prof. Keita Matsuno (Division of Risk Analysis and Management, International Institute for Zoonosis Control, Hokkaido University, Sapporo, Hokkaido, Japan) for his valuable advice on the pharmacokinetic/pharmacodynamic analysis. We also thank Ms. Mayumi Endo (Laboratory of Microbiology, Faculty of Veterinary Medicine, Hokkaido University, Sapporo, Hokkaido, Japan), Ms. Mai Tamba (One Health Research Center,

Hokkaido University, Sapporo, Hokkaido, Japan) and Dr. Yared Beyene (Laboratory of Toxicology, Department of Environmental Veterinary Science, Faculty of Veterinary Medicine, Hokkaido University, Sapporo, Hokkaido, Japan) for their technical support.

## Author contributions

**Conceptualization:** Yo Shimazu, Norikazu Isoda, Takahiro Hiono, Ryo Daniel Obara, Mariko Miki, Hiromi Osaki, Takao Shishido, Yoshihiro Sakoda.

**Data curation:** Yo Shimazu, Yoshinori Ikenaka.

**Formal analysis:** Yo Shimazu, Yoshinori Ikenaka.

**Funding acquisition:** Yo Shimazu, Yoshihiro Sakoda.

**Investigation:** Yo Shimazu, Norikazu Isoda, Takahiro Hiono, Yik Lim Hew, Daiki Kobayashi, Misato Shibazaki, Bao Linh Nguyen, Tatsuru Morita.

**Methodology:** Yo Shimazu, Norikazu Isoda, Takahiro Hiono, Ryo Daniel Obara, Mariko Miki, Hiromi Osaki, Takao Shishido, Yoshinori Ikenaka, Yoshihiro Sakoda.

**Project administration:** Yoshihiro Sakoda.

**Resources:** Yoshinori Ikenaka, Yoshihiro Sakoda.

**Supervision:** Yoshinori Ikenaka.

**Validation:** Norikazu Isoda, Takahiro Hiono, Hiromi Osaki, Takao Shishido, Yoshinori Ikenaka, Yoshihiro Sakoda.

**Visualization:** Yo Shimazu.

**Writing – original draft:** Yo Shimazu.

**Writing – review & editing:** Norikazu Isoda, Takahiro Hiono, Yik Lim Hew, Daiki Kobayashi, Misato Shibazaki, Bao Linh Nguyen, Tatsuru Morita, Ryo Daniel Obara, Mariko Miki, Hiromi Osaki, Takao Shishido, Yoshinori Ikenaka, Yoshihiro Sakoda.

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
