## [Decision Letter · Decision Letter 0]

12 Jan 2026

Dear Dr. Sakoda,

Thank you for submitting your manuscript to PLOS ONE. After careful consideration, we feel that it has merit but does not fully meet PLOS ONE’s publication criteria as it currently stands. Therefore, we invite you to submit a revised version of the manuscript that addresses the points raised during the review process.

We look forward to receiving your revised manuscript.

Kind regards,

Mohamed Samy Abousenna, Ph.D

Academic Editor

PLOS One

Journal Requirements:

“This research was conducted as part of a collaborative research project between Hokkaido University and Shionogi & Co., Ltd. It was supported by the Environment Research and Technology Development Fund [JPMEERF20254004] of Environmental Restoration and Conservation Agency provided by Ministry of the Environment of Japan. Additionally, this work was partially supported by the Japan Agency for Medical Research and Development (AMED) [grant numbers JP223fa627005 and JP24wm000125008] and by the World-Leading Innovative and Smart Education Program (1801) from the Ministry of Education, Culture, Sports, Science and Technology, Japan. It was also supported by the Japan Science and Technology Agency SPRING grant [number JPMJSP2119].”

Reviewer's Responses to Questions

**Comments to the Author**

1. Is the manuscript technically sound, and do the data support the conclusions?

Reviewer #1: Yes

Reviewer #2: Yes

2. Has the statistical analysis been performed appropriately and rigorously?

Reviewer #1: Yes

Reviewer #2: Yes

3. Have the authors made all data underlying the findings in their manuscript fully available?

Reviewer #1: Yes

Reviewer #2: Yes

4. Is the manuscript presented in an intelligible fashion and written in standard English?

Reviewer #1: Yes

Reviewer #2: Yes

Reviewer #1: The authors used duck as a model to evaluate the therapeutic efficacy of baloxavir marboxil (BXM) to H5N1 HPAIV infection. They demonstrated that the administration of BXM within two days post-infection in ducks effectively reduces mortality and virus shedding, which may contribute to the establishment of therapeutic strategies for rare avian species. The experiment was well designed and the results were convincing.

1. The authors described in the abstract as “Pharmacokinetic/pharmacodynamic analysis of BXA suggested that parameters such as Cmax and AUC0–24hr were correlated with the suppression of virus shedding.” and in the discussion as “this study found that AUC0–24hr and Cmax showed a higher correlation with the suppression of virus shedding than C24hr”. However, they described in the result as “In oral swabs, reductions in virus titers correlated with both AUC0–24hr and Cmax,┄┄ In contrast, in cloacal swabs, C24hr showed only a weak positive correlation with the reduction in virus shedding”. The statement in the abstract and the discussion are inaccurate.

2. In the preliminary test, what is the challenge route for duck infections?

Reviewer #2: This manuscript evaluates the therapeutic efficacy of baloxavir marboxil (BXM) against highly pathogenic avian influenza virus (HPAIV) infection in a duck model. Using four-week-old ducks experimentally infected with H5N1, the authors assessed the effects of BXM treatment on viral shedding and survival, informed by pharmacokinetic/pharmacodynamic analyses. The study reports that BXM administration within 48 hours post-infection significantly reduces mortality and viral shedding in infected birds. Based on these findings, the authors suggest that BXM may have potential as a therapeutic intervention for HPAIV infection in ducks and possibly other avian species of conservation concern.

This manuscript presents a scientifically sound study with an appropriate experimental design and clear presentation of results. The data support the authors’ conclusions. I recommend minor revisions to improve clarity and presentation.

Line 57:

The authors should specify the exact HPAIV lineage responsible for the global spread, namely clade 2.3.4.4b.

Line 108:

The term “rare avian species” is insufficiently defined. The authors should specify the species of interest.

Line 173:

The authors should explain why plasma concentrations of baloxavir acid (BXA) were not measured in ducks administered with 0.5 mg/kg BXM.

Line 175:

The sampling time points require clarification. Were samples collected at 2, 8, and 24 hours following the initial dose across two days post-infection (e.g., spanning 2 and 3 dpi), or were these time points collected at 2 dpi and repeated again at 3 dpi?

Line 249:

Please specify the magnitude of the difference in viral titers (e.g., in log₁₀ TCID₅₀/ml) to make this statement more quantitative.

Line 266:

As noted in the Methods, the authors should state why ducks administered 0.5 mg/kg BXM were excluded from the pharmacokinetic analysis.

Line 293:

The reported R² values (0.659 and 0.678) indicate moderate correlations between reductions in viral titers and AUC₀–₂₄h/Cmax in oral swabs. Were these correlations formally tested for statistical significance (e.g., p-values)?

**Do you want your identity to be public for this peer review?** For information about this choice, including consent withdrawal, please see our For information about this choice, including consent withdrawal, please see our Privacy Policy .

Reviewer #1: No

Reviewer #2: No

---

## [Author Response · Author response to Decision Letter 1]

17 Mar 2026

General comments on the revision of the manuscript and main changes

In response to the reviewers’ constructive comments, we have carefully revised the manuscript and performed additional analyses and experiments to improve the clarity and scientific rigor of the study. The major revisions are summarized as follows:

1. The pharmacokinetic/pharmacodynamic analysis was updated using additional experimental data, including newly obtained plasma concentration measurements at the 0.5 mg/kg dose. This updated analysis demonstrated that AUC0–24hr was positively correlated with reductions in virus shedding in oral swabs. The Abstract, Results, and Discussion sections have been revised accordingly to ensure consistency with these findings.

2. The relevant reference has been added to the revised manuscript based on the updated discussion (Lines 524–526).

3. The experimental procedures including the virus challenge route and the time points for plasma sampling were clarified.

4. The magnitude of viral titer reductions at each dose level have been included in the Results section to improve the clarity and precision of data interpretation.

5. The lineage classification of the HPAIV strains used in this study has been explicitly described, including identification of clade 2.3.4.4b as the lineage responsible for the recent global spread.

6. The terminology for the target avian species was revised throughout the manuscript to more precisely reflect the targeted avian species.

7. Additional methodological details were provided to clarify the experimental design and address technical limitations.

8. The tests of no correlation, and the corresponding results for pharmacokinetic/pharmacodynamic analysis have been added to the manuscript.

9. The relevant reference that had been inadvertently omitted has been added to the revised manuscript (Lines 165, 497–498).

10. All modifications in the revised manuscript are highlighted in yellow for reviewer #1, green for reviewer #2, blue for additional modifications from our side, and gray for modifications related to format.

Reviewer #1:

Comment1:

The authors described in the abstract as “Pharmacokinetic/pharmacodynamic analysis of BXA suggested that parameters such as Cmax and AUC0–24hr were correlated with the suppression of virus shedding.” and in the discussion as “this study found that AUC0–24hr and Cmax showed a higher correlation with the suppression of virus shedding than C24hr”. However, they described in the result as “In oral swabs, reductions in virus titers correlated with both AUC0–24hr and Cmax,┄┄ In contrast, in cloacal swabs, C24hr showed only a weak positive correlation with the reduction in virus shedding”. The statement in the abstract and the discussion are inaccurate.

Response:

We appreciate the reviewer’s comment. An additional experiment was conducted, and pharmacokinetic/pharmacodynamic analysis was performed using the updated dataset. This analysis demonstrated that AUC0–24hr was positively correlated with reductions in virus shedding in oral swabs.

Modification:

To ensure consistency with these findings, the Abstract has been revised to state that “reductions in virus shedding in oral swabs were positively correlated with BXA exposure, with AUC0–24hr showing the strongest association” (page 2, lines 42–43).

Comment2:

In the preliminary test, what is the challenge route for duck infections?

Response:

Ducks were intranasally challenged with the virus in the preliminary test.

Modification:

This information has been added to the revised manuscript as follows: “nine ducks were divided into three groups and intranasally challenged” (page 7, lines 133–134).

Reviewer #2:

Comment (line 57):

The authors should specify the exact HPAIV lineage responsible for the global spread, namely clade 2.3.4.4b.

Response:

We thank the reviewer for this suggestion. The clade responsible for the recent global spread and the clades of the virus strains used in this study have been specified in the revised manuscript.

Modification:

The manuscript has been revised as follows: “Since 2020, infections of HPAIVs belonging to clade 2.3.4.4b have been continuously reported in various regions” (page 4, lines 54–55); and “A/crow/Hokkaido/0103B065/2022 (H5N1) (Cr/Hok/22) belonging to clade 2.3.4.4b, A/black swan/Akita/1/2016 (H5N6) (Bs/Aki/16) belonging to clade 2.3.4.4e, and A/whooper swan/Hokkaido/4/2011 (H5N1) (Ws/Hok/11) belonging to clade 2.3.2.1” (page 6, lines 109–111).

Comment (line 108):

The term “rare avian species” is insufficiently defined. The authors should specify the species of interest.

Response:

We appreciate the comment. To clarify the target species are birds maintained in zoological and aquatic institutions, the expression “rare avian species” has been revised throughout the manuscript.

Modification:

The manuscript has been revised as follows: “endangered avian species” (page 4, line 58), “captive avian species” (page 2, lines 30–31, 46; page 5, lines 96, 99, 105; page 18, line 400), and “captive birds” (page 17, line 395).

Comment (line 173):

The authors should explain why plasma concentrations of baloxavir acid (BXA) were not measured in ducks administered with 0.5 mg/kg BXM.

Response:

Due to technical limitations, blood sampling at the 0.5 mg/kg dose was not performed in the initial experiment. Plasma concentrations of BXA at this dose were subsequently determined in an additional experiment.

Modification:

The corresponding methods have been described on page 8, line 171 to page 9, line 185, and the results have been presented on page 13, line 283 to page 14, line 297.

Comment (line 175):

The sampling time points require clarification. Were samples collected at 2, 8, and 24 hours following the initial dose across two days post-infection (e.g., spanning 2 and 3 dpi), or were these time points collected at 2 dpi and repeated again at 3 dpi?

Response:

Blood samples were collected at 2, 8, and 24 hours after the initial dose, spanning from 2 to 3 dpi.

Modification:

To clarify this point, the manuscript has been revised to include the expressions “spanning from 2 to 3 dpi” (page 8, line 173; page 13, line 281) and “spanning from 6 to 7 dpi” (page 9, line 174; page 13, line 282).

Comment (line 249):

Please specify the magnitude of the difference in viral titers (e.g., in log₁₀ TCID₅₀/ml) to make this statement more quantitative.

Response:

The magnitude of the reduction in viral titers has been specified in the revised manuscript : “In oral swabs, mean reductions in virus titers were 3.2, 0.8, 1.3, 0.1, –0.7 log10 TCID50/mL in 12.5, 7.3, 4.3, 2.5, and 0.5 mg/kg group, respectively.” (page 14, lines 309–310), “In cloacal swabs, mean reductions in virus titers were 1.3, 0, 0.6, 0.7, and 0.5 log10 TCID50/mL in 12.5, 7.3, 4.3, 2.5, and 0.5 mg/kg group” (page 14, lines 312–313).

Comment (line 266):

As noted in the Methods, the authors should state why ducks administered 0.5 mg/kg BXM were excluded from the pharmacokinetic analysis.

Response:

Additional experiments were conducted to perform pharmacokinetic analysis at a 0.5 mg/kg dose. The reason for not including the 0.5 mg/kg dose in the initial experiment was explained in the revised manuscript.

Modification:

This has now been clarified in the manuscript: “The 0.5 mg/kg dose was included to allow pharmacokinetic sampling at this dose level, which was not feasible in the previous experiment due to technical limitations.” (page 9, line 184–186).

Comment (line 293):

The reported R2 values (0.659 and 0.678) indicate moderate correlations between reductions in viral titers and AUC₀–₂₄h/Cmax in oral swabs. Were these correlations formally tested for statistical significance (e.g., p-values)?

Response:

The pharmacokinetic/pharmacodynamic analysis was updated to incorporate the additional experimental data, and formal tests of no correlation were performed to assess statistical significance. Significant correlations were observed for all analyses based on oral swabs, whereas no significant correlations were detected for cloacal swabs.

Modification:

R2 values and p-values from the updated pharmacokinetic/pharmacodynamic analysis have been included in the revised manuscript: “These values correlated both AUC0–24hr, Cmax, and C24hr with R2 values of 0.703 (p<0.001), 0.645 (p<0.001), and 0.581 (p<0.001), respectively (Fig 4). In cloacal swabs, mean reductions in virus titers were 1.3, 0, 0.6, 0.7, and 0.5 log10 TCID50/mL in 12.5, 7.3, 4.3, 2.5, and 0.5 mg/kg group, and R2 values with AUC0–24hr, Cmax, and C24hr were 0.099 (p=0.170), 0.037 (p=0.213), 0.182 (p=0.339), respectively.” (page 14, lines 310–314).

---

## [Decision Letter · Decision Letter 1]

31 Mar 2026

Evaluation of therapeutic efficacy of baloxavir marboxil against high pathogenicity avian influenza virus infection in duck model

PONE-D-25-65963R1

Dear Dr. Sakoda,

We’re pleased to inform you that your manuscript has been judged scientifically suitable for publication and will be formally accepted for publication once it meets all outstanding technical requirements.

Kind regards,

Mohamed Samy Abousenna, Ph.D

Academic Editor

PLOS One

Additional Editor Comments (optional):

Reviewers' comments:

Reviewer's Responses to Questions

**Comments to the Author**

Reviewer #1: (No Response)

Reviewer #2: All comments have been addressed

2. Is the manuscript technically sound, and do the data support the conclusions?

Reviewer #1: (No Response)

Reviewer #2: Yes

3. Has the statistical analysis been performed appropriately and rigorously?

Reviewer #1: (No Response)

Reviewer #2: Yes

4. Have the authors made all data underlying the findings in their manuscript fully available?

Reviewer #1: (No Response)

Reviewer #2: Yes

5. Is the manuscript presented in an intelligible fashion and written in standard English?

Reviewer #1: (No Response)

Reviewer #2: Yes

Reviewer #1: (No Response)

Reviewer #2: (No Response)

**Do you want your identity to be public for this peer review?** For information about this choice, including consent withdrawal, please see our For information about this choice, including consent withdrawal, please see our Privacy Policy .

Reviewer #1: No

Reviewer #2: No

---

## [Editor Report · Acceptance letter]

PONE-D-25-65963R1

PLOS One

Dear Dr. Sakoda,

I'm pleased to inform you that your manuscript has been deemed suitable for publication in PLOS One. Congratulations! Your manuscript is now being handed over to our production team.

Kind regards,

on behalf of

Dr. Mohamed Samy Abousenna

Academic Editor

PLOS One